# Injuries in Wheelchair Basketball Players: A Systematic Review

**DOI:** 10.3390/ijerph19105869

**Published:** 2022-05-11

**Authors:** Karina Sá, Anselmo Costa e Silva, José Gorla, Andressa Silva, Marília Magno e Silva

**Affiliations:** 1Faculty of Physical Education, State University of Campinas, Campinas 13083-851, Brazil; jigorla@uol.com.br; 2Graduate Program in Human Movement Sciences, Federal University of Pará, Belém 66075-110, Brazil; anselmocs@ufpa.br; 3School Physical Education, Physical Therapy and Occupational Therapy, Federal University of Minas Gerais, Belo Horizonte 31270-901, Brazil; andressa@demello.net.br; 4Faculty of Physical Therapy and Occupational Therapy, Federal University of Pará, Belém 66075-110, Brazil; mariliamagno@ufpa.br

**Keywords:** para athletes, sport injuries, incidence rate, prevalence

## Abstract

Background: Sports injuries have physical and psychological effects that negatively affect sports performance. Although there are data available on sports injuries in wheelchair basketball, some aspects need to be clarified, such as the location, mechanisms and risk factors for injury, which are not well described due to variations and/or a lack of definition of injury. The aim of this study was to determine epidemiological information, primary injury characteristics and affected body regions in wheelchair basketball players; Methods: The PubMed, Science Direct, Scopus, Web of Science and Google Scholar databases were used; Results: eight articles were included in this review. The shoulder was the body region most affected (N = 60; 22.1%). When divided by body segments, the upper limbs were the most affected (N = 128; 47.2%), followed by the head and/or face (N = 53; 19.5%), trunk (included spine and ribs) (N = 48; 17.8%) and lower limbs (N = 42; 15.5%); Conclusions: wheelchair basketball players suffer a large number of injuries with different characteristics that are mainly linked to biomechanics and sport. These results can be used to guide coaches in structuring training to minimize recurring injuries, in addition to assisting in the organization of medical teams in competitions.

## 1. Introduction

Wheelchair basketball (WB) is a Paralympic sport played by two teams of five players each, comprising players with physical impairments who can be allocated to eight different classes (1.0–4.5). The game proceeds at a fast pace, in which, the teams seek to score points in the opponent’s basket. This modality is popular worldwide and has been present in all editions of the Paralympic Games [1]. Due to being a contact sport and due to the sport’s mechanics, in which, frequent shoulder movements are performed (throwing, passing, chair touching), it is expected that this practice contributes to sports injuries.

Sports injuries have both physical and psychological effects that negatively affect sports performance. Once injured, an athlete may need to abstain from the activity, which may vary from days to months [2]. The longer the withdrawal period, the more common it is to observe detraining, as well as a loss of strength and agility. In addition, psychological conditions are also related to injuries, such as anxiety, stress, depression, fear of reinjury and low self-esteem [3]. Therefore, it is important to understand the mechanisms, impact and prevention of sports injuries in Paralympic sports.

A study performed at the London 2012 Paralympic Games observed a variation in the epidemiology of injuries between sports and drew attention to the need for specific longitudinal studies for each of the different modalities [4]. In London 2012, WB recorded 34 injuries, 65% of which were acute injuries, and 23% of which were overuse injuries [5]. In the Rio 2016 Paralympic Games, 4504 interventions were recorded, in which, 399 players were treated by the physiotherapy service. For this competition, eight WB players sought physical therapy, totaling 11 treatments [6], with traumatic injuries being the primary incidents [7].

Although several studies on Olympic sports have identified injuries [8,9,10], Paralympic sports still lack this same evidence. Even though data are available regarding sports injuries in WB [5,7], some aspects need to be clarified. To date, bit data, and details such as location, mechanisms and injury risk factors, are not well-described due to variations and/or a lack of injury definition. Therefore, the aim of this study was to conduct a systematic review of the literature to determine the epidemiological information, primary injury characteristics and body regions affected in WB players.

## 2. Materials and Methods

This systematic review is presented in accordance with the Preferred Reporting Items for Systematic Reviews and Meta-Analysis (PRISMA) statement [11] and was registered with the International Prospective Register of Systematic Review (PROSPERO; available at: https://www.crd.york.ac.uk/PROSPERO/) accessed on 28 May 2020 (registration number: CRD42020159566) [12]. The study question and other systemic review procedures were addressed with reference to the following PICO strategy: WB players (participants) of highly competitive levels; data collection regarding sports injuries in WB (intervention); descriptive data about injury epidemiology in WB (comparison); and main characteristics of sports injuries in WB (outcomes).

The systematic search comprised observational studies, prospective or retrospective studies and cohort studies. Studies were eligible for inclusion according to the following criteria: (i) involving WB players, (ii) papers written in English and (iii) presents numerical data of WB injuries. Studies were excluded from analysis based on the following criteria: (i) duplicate studies and (ii) studies on WB that did not address the prevalence or incidence of injuries.

English language searches of the electronic databases PubMed/Medline, ScienceDirect, Scopus, Web of Science and Google Scholar. Articles were retrieved from electronic databases using the following search strategy: “wheelchair basketball” AND “injury” OR “injuries” OR “sports injury”. In the search performed on Google Scholar, the advanced search option was used to retrieve articles with the exact phrase “wheelchair basketball” and with at least one of the following words: “injury”, “injuries” or “sports injury”. These words needed to occur in the title of the manuscript. Identified articles on the systematic search were initially checked for relevance by two independent researchers (first and third authors). Articles were selected after a sequential reading of the title and abstract, always in this order. Subsequently, the researchers reviewed the full texts of potentially eligible articles. A third researcher (second author) resolved any disagreements among reviewers regarding inclusion of the study. The references of the articles were consulted to identify possible additional studies. The articles were placed in a list in an Excel spreadsheet and the duplicates were removed.

Data extraction was performed by two independent researchers (first and third authors), supported by a third researcher (second author) when necessary. Data extracted from WB injuries included type of injuries, body region injury, injury level, year of publication, objectives, sample, gender and primary outcomes.

Study quality was assessed according to STROBE-SIIS (Sports Injury and Illness Surveillance) Statement, a checklist of items for the reporting of observational studies on injury and illness in sports, which is considered an appropriate tool to assess the methodological quality of epidemiological studies of sports injuries and illness. This tool consists of 23 items that aim to evaluate the methodological rigor of studies.

## 3. Results

### 3.1. Included Studies

Initially, screening identified 873 records in the databases. After removing duplicates, 268 studies remained that were selected for title and abstract analysis, of which, 235 were removed. Full reading was conducted for 33 articles, and the inclusion and exclusion criteria were applied. From those 33 articles, 25 did not meet the inclusion criteria and were excluded. Finally, eight studies were included in the final analysis (Figure 1), the oldest being published in 1999 and the most recent being published in 2020.

### 3.2. Quality Assessment

In general, the articles differed little in the evaluation of methodological quality. Through analysis, we identified that the articles have good methodological quality (Appendix A).

### 3.3. Study Characteristics

Five papers were classified as cross-sectional observational studies, one as a descriptive self-report, one as a prospective study and one as a survey. The primary characteristics are briefly described in Table 1. Seven hundred and fifty-three WB players were included in the eight articles, 462 of whom were men and 291 of whom were women. Of the total number of players evaluated, 274 were injured players, with 271 injuries of different etiologies.

The region of the body that presented the highest number of injuries was the shoulder (N = 60; 22.2%), followed by the head (N = 52; 19.2%), and other body regions are described in Table 2 and Figure 2. The body segment that presented the highest number of injuries was the upper limbs (N = 128; 47.2%) followed by the head and/or face (N = 53; 19.5%), trunk (spine and ribs included) (N = 48; 17.8%) and lower limbs (N = 42; 15.5%). Concussion (N = 52, 23.8%), muscle injury/contusion (N = 34; 15.5%), myalgia (N = 33; 15.1%) and pressure injuries (N = 32, 14.6%) were the primary diagnoses (Table 3).

## 4. Discussion

The aim of this review was to determine the epidemiological information, primary injury characteristics and body regions affected in WB players. The systematic review included only quantitative studies involving 753 players of both sexes. The primary results are that 274 (36.4%) players suffered injuries, with the upper limbs being most affected (47.2%), followed by the region of the head and/or face (19.5%). Regarding established diagnoses of injuries, concussion (23.8%) followed by myalgia (15.1%) were the most reported.

The International Olympic Committee, through consensus to record and report epidemiological data on injuries and illnesses in sport in 2020, defines injury and illness as follows: injury is tissue damage or other derangement of normal physical function due to participation in sports, resulting from the rapid or repetitive transfer of kinetic energy. Illness is a complaint or disorder experienced by an athlete not related to injury [21]. In this review, most articles reported their sports injury definition, considering their research focus. Therefore, we obtained a wide variety of definitions, which makes comparison between papers’ results difficult. However, Nielsen et al. [22] argue that a single, universal definition of sports injury is not necessary but that the choice should be made to seek a balance between a variety of factors and how these factors generally compete, and the authors encourage researchers to match their choice of definition with the purpose, configuration and design of the study.

### 4.1. Study Populations

The studies selected for this review included 753 WB players, 462 of whom were men and 291 of whom were women. Observing this information, we identified that it is more common for articles to be composed of male samples and that the articles often do not consider gender differences when reporting injuries.

According to a study by Derman et al. (2018), female para-athletes are at higher risk of sports injuries (IR of 11.1 (95% CI 9.7 to 12.7), *p* < 0.05) compared to male athletes (IR of 9.3 (95% CI 8.3 to 10.4)). This may be related to a condition called “female athlete triad”, which consists of low energy availability with or without disordered eating, menstrual dysfunction and low bone density that is related to the recurrent appearance of sports injuries in women [23,24]. In addition, hormonal factors, such as differences in estrogen and relaxin activity, make women more likely to experience joint instability and ligament laxity [25,26], which can lead to injuries during sports practice.

### 4.2. Sports Injury Mechanism

Considering the mechanism, sports injuries can be classified into trauma (traumatic) or overload categories. Traumatic injuries are caused by a single, specific and identifiable event. They can occur with contact (e.g., shock of the body against structures or the opponent’s body) or without contact (e.g., sprain). Overload injuries are caused by repetitive microtrauma, without the identification of a specific event causing the injury. These lesions may have either a sudden or gradual onset [27]. Among the articles reviewed here, only one clearly described the mechanism of injury. A description of this point is important to understand factors that can lead to the occurrence of injuries, in addition to helping in the prevention process.

During sports practice, athletes are exposed to traumatic and overload injuries. In the case of basketball, the biomechanics of the sport itself can influence the appearance of injuries [28]. The use of the shoulder joint in repetitive movements (propulsion, throwing and passing) can cause the appearance of injuries due to overload. In the same way, sudden changes in direction during movement on the court and the shock with other players can cause the appearance of traumatic injuries. Knowing the biomechanics of the sport helps to minimize injuries and improve sports performance [29,30].

### 4.3. Upper Limb Injuries

With the heterogeneity in the reports, many terms were found in the papers describing injuries to the upper limbs. In summary, we categorize data of injuries in the fingers, hands, wrists, forearms/arms and shoulders into a single group: upper limb injuries.

The highest frequency of WB injuries was in the upper limb, highlighting the shoulder region. These injuries are linked to repeated movements that the sport itself requires, such as the handling of the wheelchair and the biomechanics of the throw in this position [28]. In addition, the shoulder is an anatomically unstable region, being more prone to injuries [31]. In WB, the power transmission to the pitch is with the trunk, unlike conventional basketball players, where the force starts in the lower limbs.

In the literature, shoulder injuries in wheelchair sports are primary represented by shoulder impact syndrome and rotator cuff injuries [32] that generate pain, a loss of muscle strength and a decreased range of motion, resulting in changes in biomechanics and positioning, providing muscle shortening and difficulty in performing sports and daily life tasks [28]. These injuries are related to repetitive movements and force movements performed above the head [33], activities that are present in WB practice. In addition to the shoulder region, we also highlighted injuries to the fingers, hands and wrists, which are primarily represented by fractures and sprains. In general, basketball is a contact sport and therefore promotes the appearance of these injuries, which represents a negative impact on the athlete primarily because it affects the dexterity and skill that an athlete needs to master the ball and perform movements [34].

### 4.4. Head Injuries

The primary head injury observed in these studies was concussion. Sport-related concussion is a traumatic brain injury induced by biomechanical forces and may be caused by either a direct blow to the head, face, neck or elsewhere on the body with an impulsive force transmitted to the head. Symptoms of neurological impairment usually appear quickly and resolve spontaneously; however, these signs and symptoms in some cases can appear over a few minutes to hours. The acute clinical signs and symptoms largely reflect a functional disturbance rather than a structural injury; therefore, no abnormalities were observed in standard structural neuroimaging exams [35,36].

The signs and symptoms of concussion include loss or not of consciousness, memory impairment, headache, nausea and vomiting, visual disturbances and eye movement, balance impairment and behavioral changes. However, signs and symptoms are not, by themselves, a diagnosis of concussion, and, for suspected diagnosis of concussion, the clinical signs and symptoms cannot be explained by drugs, alcohol, medication use, other injuries or other comorbidities [37,38]. At present, it is known that most athletes are able to recover from clinical symptoms, even in the first month after the injury, but the return to sports needs to occur gradually [36]. As previously mentioned, basketball is a contact sport, and, for this reason, the incidence rates of concussions in this sport are higher compared to low contact sports.

Intervention protocols and behaviors already exist in the literature, such as The Sports Concussion Assessment Tool 5 (SCAT5), which can be used on and off the court [37]. In addition, we emphasize that concussion prevention strategies, such as using specific equipment and changing sport-specific rules to avoid more serious contact, should be carefully considered.

### 4.5. Lower Limbs Injuries (Knee, Hip and Ischiatic Region)

Injuries with higher incidence found in the lower limbs included pressure sore injuries, contusions and abrasions on the skin. The appearance of pressure sore injuries is observed in players who depend on the wheelchair for their locomotion and who present sensitivity changes in areas that remain in contact with the chair, primarily ischiatic and sacrum regions; therefore, people with spinal cord injury are the most affected [16,17]. It is important to note that pressure injuries are not exclusive to athletes, but the practice of sports can be a factor that promotes their occurrence. Players with lower classifications usually have greater trunk instability and might be at risk for the occurrence of pressure injuries compared to players of higher classes who, in turn, have better postural control, since these players with lower scores do not experience posture changes, such as tilting the trunk and lowering the pressure points when sitting. These injuries present as a risk factor for poor blood circulation in the region with greater contact with the chair, pressure at specific points for long periods during the day and friction of the skin, and, within sports practice, sweat favors an environment conducive to the development of these injuries. The authors report that, if untreated, these injuries can lead to serious conditions, such as sepsis, and represent the risk of suspension from sports practice until the injury is completely healed [17].

Regarding bruises and abrasions on the skin, as has been previously indicated, WB is a contact sport where these situations can occur during the game. In certain movements, parts of the metal structure encounter the opponent’s body, which may cause these injuries.

### 4.6. Spinal Injuries (Cervical, Thoracic and Lumbar)

Spinal injuries do not seem to be directly related to sports practice, yet the incidence of pain in the spine region, primarily lumbar, in permanent wheelchair users is high, and this population seems to be more susceptible to the onset of this type of pain compared to the general population [38]. These pain symptoms, both acute and chronic, may be related to the ergonomic characteristics of the chair, since these users spend most of their time sitting in these chairs that might not have the necessary anatomical adjustments, resulting in pain [39]. In addition to ergonomic factors, it is also necessary to mention factors such as non-physical activity, muscle inactivity and neuropathies [38]. The prevention of this condition is important because pain negatively affects the quality of life of individuals, and the changes in positioning, the practice of physical activities and the realization of necessary ergonomic adjustments in the chair would be preventive factors for the onset of pain.

### 4.7. Collection Pattern

Most studies presented here do not clearly describe the characteristics of the sport injuries. Some studies that focused on different sport modalities did not individually report the number, region or diagnosis of injuries for each sport. Some of the injuries mentioned in the articles have not had their etiology or mechanism of injury studied, making it difficult to analyze whether the injury was acquired during training and games or off the court, whether the injury is prior to sports practice or not and with what mechanism. Not having this information prevents us from making coherent decisions to prevent sports injuries. Most articles defined the concepts of injuries in their publications (Table 1).

These differences demonstrate the importance of standardizing collections for studies on sports injuries. To date, there is no protocol for collecting data on injuries in wheelchair basketball, demonstrating the need to create a standard model. Clear exposure of these points is needed so that sports injury epidemiology, focusing on incidence and prevalence, have a positive impact on the literature and clinical practice [30].

In this sense, Magno and Silva [40] proposed the Sports Injury Protocol in Paralympic Sports, a sports injury data protocol in Paralympic sports that is multimodal, multi-handicap, multifactorial and accessible. This system consists of six stages: consent form, impairment data, modality data, training diary, competition diary and sports injury. Through the steps of this system, it is possible to obtain data on the characteristics of sports injuries and their internal and external factors. Determining these aspects is essential for structuring a good research methodology for sports injuries.

As previously mentioned, studies with a clear methodology enable the replication of studies and the application of results on injuries in different contexts, such as training, rehabilitation and logistics organization in sports championships.

### 4.8. Future Perspective

Within sports medicine, our findings contribute to the decision making of the sports team (doctors, physiotherapists, coaches, physical trainers) through the knowledge of the main injuries, where it is possible to structure training to prevent them. In this sense, the creation of an injury prevention protocol for the sport is interesting.

We have a model of success in football using the FIFA 11+ prevention protocol. Studies that evaluated the FIFA 11+ program found that the application of the protocol reduced the number of injured athletes and improved performance components, both neuromuscular and motor [41,42].

Currently, there is no protocol for the prevention of injuries in wheelchair basketball. Following the injuries highlighted in this review, a protocol aimed at this sport would have a pressure injury prevention program (this type of injury impacts training and can keep the athlete away from sports) and a sequence of exercises that simulate the sports practice aiming to improve movements and strengthen the upper limbs and core, with increasing difficulties, which could be used to warm up athletes. Strategies in this sense would decrease the risk of injury and, when there was an injury, decrease the time away from the athletes.

## 5. Conclusions

WB players experience several injuries with different etiologies. The largest number of injuries was observed in the upper limbs, especially in the shoulder region. An important factor in the injury process in this sport is the fact that the activity offers many contacts for its players. In this study, we determined that the most affected body regions were the shoulder, hand, head and spine. In addition, the primary diagnoses were concussions, muscle injury/contusion and myalgia. These findings can be used to direct coaches to structure training aimed at minimizing recurrent injuries in addition to helping to organize medical teams in competitions, given that sports injuries occur with greater frequency in WB.

## Figures and Tables

**Figure 1 ijerph-19-05869-f001:**
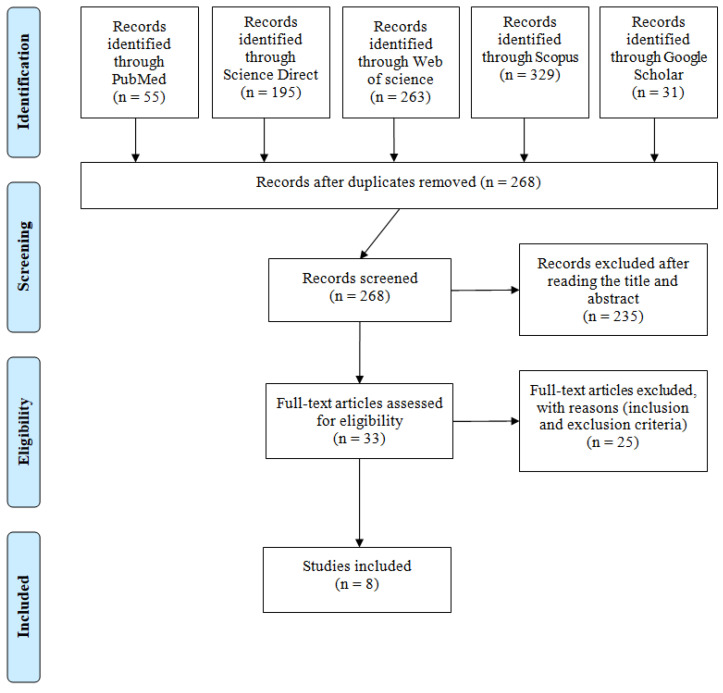
PRISMA flow diagram.

**Figure 2 ijerph-19-05869-f002:**
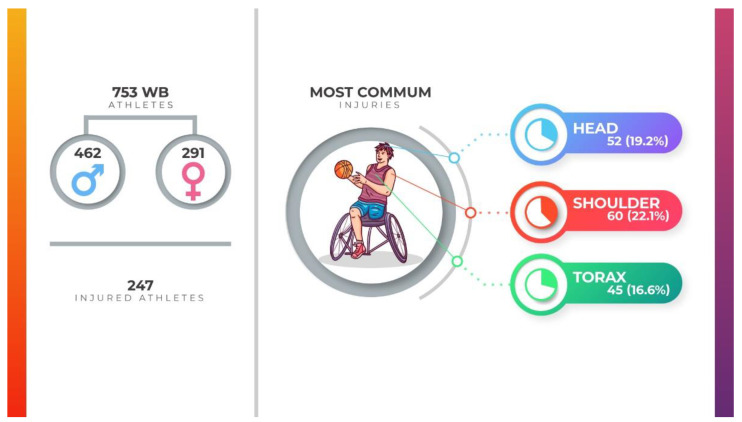
Infographic of main wheelchair basketball injuries.

**Table 1 ijerph-19-05869-t001:** Description of articles.

Author	Design	Purpose	Definition of Injury	Sample	Number of Injured Athletes	Incidence and Prevalence
Curtis and Black (1999) [13]	Descriptive self-report survey.	To assess activity level, medical history and prevalence and intensity of shoulder and upper extremity pain during functional activities in female wheelchair athletes.	Not reported.	46 WB female athletes.	33 athletes.	14% of the subjects reported shoulder pain prior to wheelchair use. 72% of the subjects reported shoulder pain since wheelchair use. 52% reported current shoulder pain.
Rocco and Saito (2006) [14]	Cross-sectional study.	To identify the most frequent sports injuries of basketball wheelchair players.	“contusion (injury caused by a direct trauma on the body leading to internal involvement); muscular rupture (solution of muscle continuity); muscle stretching (micro-lesion due to excessive stretching of the muscle); muscular cramps (muscular contractions in which the athlete cannot relax the muscle voluntarily); sprains (abrupt movement beyond the normal amplitude); joint dislocation (loss of joint congruence); fracture (solution of bone continuity), tendonitis and bursitis, among others.”	26 male WB athletes	20 athletes	54% of athletes reported pain and 79% localized in the upper limbs.
Wessels et al. (2012) [15]	Survey.	To estimate the incidence rate of WB concussion.	“Concussions are a mild traumatic brain injury (mTBI) seen in athletic participation.”	263 WB players (188 male and 75 female).	50 athletes affected in the current or previous season.	6.1% reported experiencing a concussion during the publication year season
Mutsuzaki et al. (2014) [16]	Cross-sectional study.	To use ultrasound to investigate tissue injuries in male WB players and determine factors associated with injuries.	“Deep tissue injury is defined as injury to soft tissue resulting from pressure and/or shear.”	20 WB male athletes.	9 athletes.	45% of players had low-echoic lesions
Shimizu et al. (2017) [17]	Cross-sectional study.	To investigate deep tissue injuries (DTIs) in elite WB players and identify factors associated with their occurrence.	“Deep tissue injury was defined as a purple or maroon localized area of discoloured intact skin or a blood-filled blister due to damage to the underlying soft tissue from pressure and/or shear forces.”	22 female WB athletes.	15 athletes.	68.2% of players reported DTIs
Huzmeli et al. (2017) [18]	Cross-sectional study.	To determine the prevalence and nature of injuries in wheelchair sports participants.	Not reported.	15 WB athletes (14 male and 1 female).	4 athletes in the last 12 months.	26.6% of individuals had injuries in the past one year and 75% of them had injuries because of muscle tear.
Soo Hoo et al. (2018) [19]	Descriptive cross-sectional study.	To evaluate the demographics, training regime and injuries suffered by para-athletes participating in sports clubs and to evaluate the type of medical care of athletes and the prevalence of those with spasticity.	“an injury while playing an adaptive sport that required you to sit out of a practice or a game.”	43 athletes, of which, 25 are WB players (22 male and 3 female).	11 players in the last 12 months.	In the past 12 months, 39.5% of athletes surveyed sustained an injury. Injury prevalence by sport was 44% in WB.
Hollander et al. (2020) [20]	Prospective study	To assess the rate and characteristics of injuries during the WB World Championships 2018 (WBWC).	“any newly incurred musculoskeletal complaint (traumatic or overuse) and/or concussion during the tournament receiving medical attention regardless of the consequences for participation.”	336 players (male: 192; female: 144)	132 players	75.8 per 100 players (95% CI: 60.9–90.7) or 68.9 per 1000 player-days (55.4–82.4).

**Table 2 ijerph-19-05869-t002:** Summary of injuries by body.

**Body Regions**	**Number of Injuries (%)**	**Studies**
Shoulder	60 (22.2%)	[13,14,18,19,20]
Spine (cervical/thoracic/lumbar)	45 (16.6%)	[14,18,19,20]
Head	52 (19.2%)	[15,19,20]
Wrist	17 (6.3%)	[14,19,20]
Elbow	18 (6.6%)	[19,20]
Sacrum	21 (7.7%)	[16,19,20]
Arm	11 (4.1%)	[14,18,20]
Ischiatic region	15 (5.5%)	[16,17]
Hand/fingers	19 (7.0%)	[14,18,19,20]
Forearm	3 (1.1%)	[18]
Knee	4 (1.5%)	[18,19,20]
Face	1 (0.4%)	[19]
Ribs	1 (0.4%)	[19]
Abdomen	2 (0.7%)	[20]
Thigh	2 (0.7%)	[20]
**Body Segments**	**Number of Injuries (%)**	**Studies**
Upper limb	128 (47.2%)	[13,14,18,19]
Trunk (spine and ribs included)	48 (17.8%)	[14,18,19]
Lower limbs	42 (15.5%)	[16,17,18,19]
Head and/or face	53 (19.5%)	[15,19]

**Table 3 ijerph-19-05869-t003:** Diagnosis of injuries.

Diagnosis	N (%)	Studies
Concussions	52 (23.8%)	[15,19]
Muscle injury/contusion	34 (15.5%)	[14,18,19]
Myalgia	33 (15.1%)	[18]
Pressure injury	32 (14.6%)	[16,17]
Sprain and fracture	25 (11.4%)	[14,18,19,20]
Muscle spasm	25 (11.4%)	[20]
Tendinopathies	13 (5.9%)	[14,20]
Impingement	5 (2.3%)	[20]

## Data Availability

Not applicable.

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
