# Peer review of "Injuries in Wheelchair Basketball Players: A Systematic Review"

_ijerph, 2022, doi:10.3390/ijerph19105869_

Round 1

Reviewer 1 Report

The aim of this study was to conduct a systematic review of the literature to determine the epidemiological information, primary injury characteristics and body regions affected in weelchair basketball players.

Paralympic sport is a sport of increasing research and scientific interest. This is an interesting review that can help sport professionals to improve their physical preparations.

Here are my contributions:

- Line 41, the difference between acute injuries and acute or chronic injuries is not distinguished, as both groups include acute injuries.

  • Table 1 has a lot of content (words) and is neither practical nor easy to read. Please summarise its content as much as possible.
  • Table 1, Add the author(s) and then the reference.
  • Line 142-174, this section does not provide information relevant to the subject of the review.
  • Line 175, again this section does not contribute information of interest to the review. This section could be added to any review related to sports injuries.
  • Line 250, taking into account the high rate of head injuries, couldn't some kind of recommendation be made as in other sports where the head can suffer some kind of impact?
  • Line 315, two authors and the year of publication are added, but if the reference is added then ….
  • Línea 346, corregir “etiologías"

Author Response

Dear reviewer,

Thank you for working on reviewing this manuscript.

The following is a point-by-point approach to the suggested revisions:

Point 1: Line 41, the difference between acute injuries and acute or chronic injuries is not distinguished, as both groups include acute injuries.

Response 1: We have re-reviewed the cited information. We had described these findings in the same way as discussed in the study that provided this information. To avoid confusion, we chose to describe only information on acute and overuse injuries.

Point 2: Table 1 has a lot of content (words) and is neither practical nor easy to read. Please summarise its content as much as possible.

Response 2: We tried to make it as clear as possible.

Point 3: Table 1, Add the author(s) and then the reference.

Response 3: We add the authors and year of publication before the reference.

Point 4: Line 142-174, this section does not provide information relevant to the subject of the review.

Response 4: We accept the suggestion and withdraw this section.

Point 5: Line 175, again this section does not contribute information of interest to the review. This section could be added to any review related to sports injuries.

Response 5: We accepted the suggestion and removed most of it but left what we felt was necessary.

Point 6: Line 250, taking into account the high rate of head injuries, couldn't some kind of recommendation be made as in other sports where the head can suffer some kind of impact?

Response 6: we took your suggestion into account and added a new paragraph to this section that indicates protocols and interventions that are already described in the literature.

Point 5: Line 315, two authors and the year of publication are added, but if the reference is added then ….

Response 5: We accepted the suggestion and kept only the reference, removing the name of the authors.

Point 6: Línea 346, corregir “etiologías"

Response 6: We corrected the spelling of the word.

Reviewer 2 Report

The authors present an interesting manuscript. However, there are certain methodological and content aspects that must be carefully reviewed so that the manuscript should be considered for publication in this prestigious journal.

The conclusions of the abstract are too ambiguous, they should be specific: "wheelchair basketball players suffer a large number of injuries with different characteristics that are mainly linked to biomechanics and sport".

In the introduction, more detailed information on this sport is missing to contextualize the factors that are going to be analyzed, as well as for the harmful epidemiology.

The information of the actions by the authors specified in materials and methods is irrelevant for the study, especially in this section.

Articles that are not full text should not be excluded from the study as they may contain relevant information to take into account for the study to be relevant enough to be published in this prestigious journal.

They must specify what it means: "presents quantitative data of WB injuries"

"Studies were excluded from analysis based on the following 66 criteria: (i) duplicate studies and (ii) other type of studies involving WB". What does other type of studies mean? All exclusion and inclusion criteria must be detailed in detail.

Only anatomical and diagnostic regions are specified. What about the moment, mode, type,... and other relevant information on the types of injury? This information is important if we are talking about epidemiology, not lesion characteristics.

The results reflect the total sum of injuries in general terms where the base of athletes is different and where each one derives from different studies. Characteristics differentiated by studies are not taken into account. The results should be expressed differently separated by articles. Only if in one of the studies a percentage and the n have been very high, it will bias the total sample. What scientific basis does this count have been made? Is this methodology correct?

There is too much information included in the discussion that should not be in this section. Historical data, search criteria, n of results,... this information must be moved to its corresponding sections. The discussion has to be to the point. Compare the results of the articles and perform the interpretation of the similarities, differences and justifications to draw clear conclusions. Always follow this order of factors in order to make reading easier for readers; the aim of this study is to determine epidemiological information.

According to the end of his introduction: "Even though data is available regarding sports injuries in WB [5, 7], some aspects need to be clarified. To date, bit data, and details such as location, mechanisms, and injury risk factors, are not well-described due to variations and/or a lack of injury definition.Therefore, the aim of this study was to conduct a systematic review of the literature to determine the epidemiological information, primary injury characteristics and body regions affected in WB players. "
The results and the discussion must be structured in relation to the factors that they want to respond to. In addition, the conclusions, just like in the abstract, must be clear, not general.

Author Response

Dear reviewer,

Thank you for working on revising this manuscript.

The following is a point-by-point approach to the suggested revisions:

Point 1: The conclusions of the abstract are too ambiguous, they should be specific: "wheelchair basketball players suffer a large number of injuries with different characteristics that are mainly linked to biomechanics and sport".

Response 1: we accepted the suggestion and reworded the sentence.

Point 2: In the introduction, more detailed information on this sport is missing to contextualize the factors that are going to be analyzed, as well as for the harmful epidemiology.

Response 2: In the introduction text, we had already described that wheelchair basketball is a contact sport and due to this specific characteristic, athletes would be subject to injuries. In this way, we have added some game mechanics moves that can influence the appearance of injuries.

Point 3: The information of the actions by the authors specified in materials and methods is irrelevant for the study, especially in this section.

Response 3: To write materials and methods, we use the Preferred Reporting Items for Systematic Reviews and Meta-Analyses (PRISMA) statement. PRISMA is a minimal evidence-based item set for reporting in systematic reviews and meta-analyses, which is used as the basis for reporting systematic reviews. In this way we follow these criteria and describe them in the materials and methods as has been done in review works, thus making the work replicated if necessary.

Point 4: Articles that are not full text should not be excluded from the study as they may contain relevant information to take into account for the study to be relevant enough to be published in this prestigious journal.

Response 4: To avoid bias and following the PRISMA recommendations, we prefer to include only original and complete studies, thus excluding abstracts and book chapters.

Point 5: They must specify what it means: "presents quantitative data of WB injuries"

Response 5: To make it clearer, we replaced the term "quantitative" with "numerical".

Point 6: "Studies were excluded from analysis based on the following 66 criteria: (i) duplicate studies and (ii) other type of studies involving WB". What does other type of studies mean? All exclusion and inclusion criteria must be detailed in detail.

Response 6: We reworded the suggested sentence to make the text clearer.

Point 7: Only anatomical and diagnostic regions are specified. What about the moment, mode, type... and other relevant information on the types of injury? This information is important if we are talking about epidemiology, not lesion characteristics.

Response 7: We did not add this information to the results, as described in our discussion, only one study clearly defined the mechanism of injury. Therefore, in our discussion, we also raised the importance of clearly reporting these points in prevalence and incidence studies.

Point 8: The results reflect the total sum of injuries in general terms where the base of athletes is different and where each one derives from different studies. Characteristics differentiated by studies are not taken into account. The results should be expressed differently separated by articles. Only if in one of the studies a percentage and the n have been very high, it will bias the total sample. What scientific basis does this count have been made? Is this methodology correct?

Response 8: The individual results of each study are described in table 1. The percentages of our results took into account the sum of the n of all articles, thus demonstrating the general characteristics, as in systematic reviews.

Point 9: There is too much information included in the discussion that should not be in this section. Historical data, search criteria, n of results... this information must be moved to its corresponding sections. The discussion has to be to the point. Compare the results of the articles and perform the interpretation of the similarities, differences and justifications to draw clear conclusions. Always follow this order of factors in order to make reading easier for readers; the aim of this study is to determine epidemiological information.

Response 9: We accept the suggestion and to make the discussion clearer and more objective, we have removed some information that we believe to be unnecessary for the purpose of our research.

Point 10: According to the end of his introduction: "Even though data is available regarding sports injuries in WB [5, 7], some aspects need to be clarified. To date, bit data, and details such as location, mechanisms, and injury risk factors, are not well-described due to variations and/or a lack of injury definition. Therefore, the aim of this study was to conduct a systematic review of the literature to determine the epidemiological information, primary injury characteristics and body regions affected in WB players. "

The results and the discussion must be structured in relation to the factors that they want to respond to. In addition, the conclusions, just like in the abstract, must be clear, not general.

Response 10: We structured the discussion according to the results, so that everything that was found could be properly discussed and thus meet the objective of this systematic review. Thus, we have removed from the text some unnecessary information for this work.

Reviewer 3 Report

Dear authors,

Your research is valuable in terms of its subject, scope, and content. It is also nice to have wheelchair basketball players. Overall, the research is well-written and fluent. Method, results, and discussion are written in a fluent language and sufficient literature is used. There are some minor corrections that I would like you to add to the introduction only, after these corrections your article is eligible for publication. Finally, I recommend that you make a minor language revision to your article.

- In the introduction, the authors only included the literature related to the Olympic games. However, not only in the research but also in the information about the injuries that may occur due to the structure of the wheelchair, it would be appropriate to support it with references, if any.

Author Response

Dear,

Thank you for working on revising this manuscript.

The following is a point-by-point approach to the suggested revisions:

Point 1: In the introduction, the authors only included the literature related to the Olympic games. However, not only in the research but also in the information about the injuries that may occur due to the structure of the wheelchair, it would be appropriate to support it with references, if any.

Response 1: In our introduction, we used injury data that were collected during the 2012 and 2016 Paralympic games. We mention that there are many studies on Olympic basketball, but still, few studies regarding wheelchair basketball.

We investigated if there is any document that related injuries to the structure of the chair, however, we were not successful.

Additionally, we performed a language review throughout the text.

Round 2

Reviewer 1 Report

Congratulations to the authors. The changes have been considerable and greatly improved the manuscript.

Reviewer 2 Report

After reviewing the responses to the suggestions made, the authors have answered appropriately and made the pertinent changes. Therefore I have no objection to the publication of this manuscript.